# Unsupervised Representation Learning by Invariance Propagation

**Feng Wang, Huaping Liu,** * **Di Guo, Fuchun Sun**
Department of Computer Science and Technology, Tsinghua University, China
Beijing National Research Center for Information Science and Technology
`wang-f20@mails.tsinghua.edu.cn,hpliu@tsinghua.edu.cn`
`guodi.gd@gmail.com,fcsun@tsinghua.edu.cn`

## Abstract

Unsupervised learning methods based on contrastive learning have drawn increasing attention and achieved promising results. Most of them aim to learn representations invariant to instance-level variations, which are provided by different views of the same instance. In this paper, we propose Invariance Propagation to focus on learning representations invariant to category-level variations, which are provided by different instances from the same category. Our method recursively discovers semantically consistent samples residing in the same high-density regions in representation space. We demonstrate a hard sampling strategy to concentrate on maximizing the agreement between the anchor sample and its hard positive samples, which provide more intra-class variations to help capture more abstract invariance. As a result, with a ResNet-50 as the backbone, our method achieves 71.3% top-1 accuracy on ImageNet linear classification and 78.2% top-5 accuracy fine-tuning on only 1% labels, surpassing previous results. We also achieve state-of-the-art performance on other downstream tasks, including linear classification on Places205 and Pascal VOC, and transfer learning on small scale datasets.

## 1 Introduction

Deep convolutional neural networks have gained great progress since the emergence of large-scale annotated datasets [7, 44]. The learned networks perform well on classification tasks [23, 37, 18], and can be transfered well to other tasks such as detection [13, 36] and segmentation [27, 17]. However, such progress requires large-scale human-annotated datasets, which are difficult to acquire. Unsupervised learning is introduced to give us the promise to learn useful representations without manual annotations. Specifically, many self-supervised methods are proposed to learn representations by solving handcrafted auxiliary tasks, such as jigsaw puzzle [31], rotation [12], colorization [42], etc. Although the self-supervised methods have gained remarkable performance, the design of pretext tasks depends on the domain-specific knowledge, limiting the generality of both the learned representations and the design of future methods. Recently, methods based on contrastive learning [39, 33, 38, 16, 1] have drawn increasing attention and achieved promising results. Most of them aim to learn representations invariant to different views of the same instance, such as data augmentations [39, 33, 16, 1], color information [38] and context information [33], while ignoring the relations between different instances which are the key to reflect the global semantic structure.

In this work, we present Invariance Propagation (InvP), a novel method which embodies the relations between different instances to learn representations with category-level invariance. Specifically, InvP discovers positive samples by recursively propagating the local invariance through the k-nearest neighbors graph. We keep k small to preserve a high semantic consistency. By applying transitivity

---

on the kNN graph, we can obtain positive images exhibiting richer intra-class variations. In this way, the positive samples contain different semantically consistent instances discovered by our algorithm, which enable the network to learn representations invariant to intra-class inter-instance variations. InvP keeps the positive samples residing in the same high-density regions, making the positive samples more consistent, which coincides with the smoothness assumption [4] widely adopted in semi-supervised learning.

After obtaining the positive samples, our goal is to learn a model which maps the pixel space to an embedding space where positive samples are attracted and negative samples are separated. To learn the model effectively, we demonstrate a hard sampling strategy. Specifically, we regard those positive samples with low similarities to the anchor sample as hard positive samples, and concentrate on maximizing the agreement between the anchor sample and its hard positive samples. We will show in the experiments section that the hard positive samples provide more intra-class variations, which helps improve the robustness of the learned representations.

We evaluate our method on extensive downstream tasks including linear classification on ImageNet [7], Places205 [44] and Pascal VOC [10], transfer learning on seven small scale datasets, semi-supervised learning and object detection. Overall, our contributions can be summarized as follows:

- We propose Invariance Propagation, a novel unsupervised learning method that exploits the relations between different instances to learn representations invariant to category-level variations. Our method is both effective and reproducible on standard hardware.

- We demonstrate a hard sampling strategy to find positive samples that provide more intra-class variations to help capture more abstract invariance. Experiments are conducted to validate the effectiveness of the proposed hard sampling strategy.

- We conduct extensive quantitative experiments to validate the effectiveness of our method, achieving competitive results on object detection and state-of-the-art results on ImageNet, Places205, and VOC2007 linear classification, semi-supervised learning, and transfer learning.

- We conduct qualitative experiments, including the visualization of similarity distribution, which shows our method successfully captures the category-level invariance, and the visualization of hard positive samples, which gives an intuitive understanding of the hard sampling strategy.

## 2   Related Works

**Self-supervised Learning.** Many self-supervised learning methods are proposed to solve artificially designed pretext tasks. The underlying assumption is that solving the pretext tasks learns some general knowledge, which is also required by some downstream tasks. Examples include context prediction [8], jigsaw puzzle [31], rotations [12], colorization [42, 24], context encoder [35], split brain [43], learning by count [32], etc. Although the self-supervised learning methods have gained remarkable performance, the design of pretext tasks depends on the domain-specific knowledge, which limits the generality of both the learned representations and the design of future methods. A more general paradigm of self-supervised learning is to utilize the clustering technique. Deep cluster [2, 3] first proposes to use clustering results as pseudo-labels to help learn useful representations. Very recently, BoWNet [11] is proposed to learn representations by classifing the visual words produced by clustering, achieving competitive results on many downstream tasks.

**Contrastive Learning.** Recently, unsupervised methods based on contrastive loss have drawn increasing attention and achieved state-of-the-art performances. These methods use contrastive loss to learn representations which are invariant to data augmentations [1, 39, 16], context information [33, 19], cross-channel information [38] or different pretext tasks [28]. Methods based on contrastive learning require many negative samples, and some works concentrate on the way to store negative samples. Wu et al. [39] first propose to save the computed features to a memory bank. He et al. [16] propose MoCo to retrieve negative samples from a momentum queue. There are also some works [5, 40] directly use features of the current batch as negative samples. Several works have tried to learn representations invariant to inter-instance variations. Huang et al. [20] propose neighborhood discovery to learn more robust representations by incorporating the nearest neighbors to positive samples, and they adopt a curriculum manner to choose positive samples progressively. Zhuang et al. [46] propose local aggregation to incorporate the samples in the same cluster into positive samples,

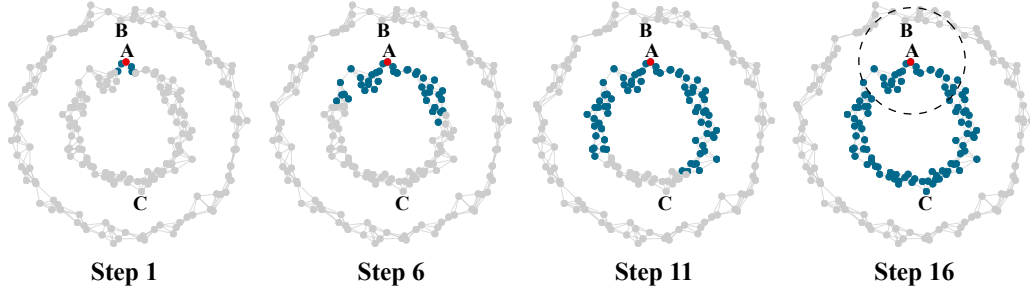

Figure 1: Illustration of the positive sample discovery algorithm. We construct the kNN graph to find the positive samples of point A recursively (k=6 in the example). The dark blue points represent the discovered positive samples. We show the positive samples of A in step 1,6,11 and 16. In each step, all $k$-nearest neighbors of the current positive samples are added to the positive sample set.

and they use nearest neighbors as background samples. With regard to concurrent work, PCL [26] also concentrates on the inter-instance relations and combines the contrastive loss with clustering.

## 3  Invariance Propagation

Given an unlabled dataset $X = \{x_1, ..., x_n\}$, our goal is to learn an embedding function $f_\theta$ that maps $X$ to $V = \{v_1, ..., v_n\}$ where $v_i = f_\theta(x_i)$ resides in a low dimensional manifold in which semantically consistent images are concentrated and semantically inconsistent images are separated. To model the relative similarity of two samples, we follow [39, 20, 46] to define the probability of sample $v_i$ being recognized as the $j$-th sample as:

$$P_{v_i}(j) = \frac{\exp(\bar{v}_j \cdot v_i / \tau)}{\sum_{k=1}^{n} \exp(\bar{v}_k \cdot v_i / \tau)} \tag{1}$$

where $\tau$ is the temperature parameter and $\bar{v}_k$ may come from memory bank [39], momentum queue [16], or the current batch of features [40, 5]. In this paper, we simply use the memory bank to update $\bar{v}_i$ as an exponential moving average of $v_i$. Furthermore, given an image set $\mathcal{S}$, we then define the probability of $v_i$ being recognized as an image in $\mathcal{S}$ as:

$$P_{v_i}(\mathcal{S}) = \sum_{j \in \mathcal{S}} P_{v_i}(j) \tag{2}$$

which is similar to [20, 46]. Next, with these definitions, we will introduce the positive sample discovery algorithm to find the semantically consistent samples for each image and then introduce the hard sampling strategy to learn our models effectively.

### 3.1  Positive Sample Discovery

Formally, we define the indices of the $k$-nearest neighbors of feature $v_i$ as $\mathcal{N}_k(i)$. Correspondingly, we denote $\mathcal{N}_k(\boldsymbol{I})$ as the union set of $\mathcal{N}_k(i)$ for $i \in \boldsymbol{I}$. Note that $i \notin \mathcal{N}_k(i)$. After giving these definitions, we formulate the positive sample set $\mathcal{N}(i)$ of image $x_i$ as:

$$\mathcal{N}(i) = \mathcal{N}_k(i) \cup \mathcal{N}_k(\mathcal{N}_k(i)) \cup ... \cup \underbrace{\mathcal{N}_k(\mathcal{N}_k(\mathcal{N}_k(...\mathcal{N}_k(i))))}_{l} \tag{3}$$

The process is illustrated in Fig 1. In each step, all $k$-nearest neighbors of the current discovered positive samples are added to the positive sample set. The process repeats $l$ steps. In another view, the positive samples discovered by our algorithm can be regarded as those samples whose graph distances from $v_i$ are less or equal than $l$.

The underlying principle of the positive sample discovery algorithm is the smoothness assumption [4]. Specifically, if two points $v_1$ and $v_2$ in a high-density region are close, then their semantic information should be similar. By transitivity, the assumption implies that if two samples are linked by a path of high density, their semantic information is likely to be close. In our method, we keep $k$ small to

guarantee the high-density condition and adjust $l$ to find the appropriate number of positive samples. This can be illustrated as Fig 1, in which we can observe the discovered positive samples all reside in a connected high-density region.

We compare our algorithm with the KNN algorithm [20], i.e., choosing the K-nearest neighbors of $v_i$ as positive samples (To distinguish the KNN here with the kNN adopted by our method, we use the uppercase K here, while the lowercase k in our method). In the last step of Fig 1, samples in the dashed circle are positive samples discovered by KNN. By comparison, we observe that point B is included in K-nearest neighbors of point A and point C is not included. However, point A and B do not belong to the same category because they reside in different high-density regions. In a good embedding space, samples with the same category should not be separated by a low-density region. The KNN method uses Euclidean distance as the global metric, which is not suitable for the manifold structure. As a comparison, our method uses Euclidean distance as the local metric to construct the kNN graph (manifolds locally have the structure of Euclidean space), and we uses graph distance as the global metric.

## 3.2 Hard Sampling Strategy

Up to now, we can learn the unsupervised model by optimizing $-\log(P_{v_i}(\mathcal{N}(i)))$ for all $v_i$. However, if we simply optimize the loss, the penalty strength on $P_{v_i}(j)$ for all $j \in \mathcal{N}(i)$ is equal. For example, if $P_{v_i}(j_1)$ is sufficiently large and $P_{v_i}(j_2)$ is relative small ($j_1, j_2 \in \mathcal{N}(i)$), the gradients with respect to $P_{v_i}(j_1)$ and $P_{v_i}(j_2)$ are of same amplitudes, which will result in the situation that optimizing $-\log(P_{v_i}(\mathcal{N}(i)))$ tends to maximize some easy optimized similarities. Ideally, $v_i$ should be similar to all discovered positive samples, instead of only some of them, to capture abstract invariance effectively.

To solve the problem, we maximize the infimum of $\{P_{v_i}(j)|j \in \mathcal{N}(i)\}$, i.e., the minimum probability, to maximize all probabilities. In practice, we select $P$ samples with the lowest similarity to construct the hard positive sample set $\mathcal{N}^h(i)$. These hard positive samples deviate far from the anchor sample such that they provide more intra-class variations, which is beneficial to learn more abstract invariance.

Correspondingly, we also choose the hard negative samples. We denote the $M$ nearest neighbors of $v_i$ as $\mathcal{N}_M(i)$. The $M$ is large enough such that $\mathcal{N}(i) \subseteq \mathcal{N}_M(i)$. Then we denote the hard negative sample set $\mathcal{N}_{neg}(i) = \mathcal{N}_M(i) - \mathcal{N}(i)$, and the background set $B(i) = N_{neg}(i) \cup \mathcal{N}^h(i)$. After giving these definitions, we formulate the loss for $x_i$ as follows:

$$\mathcal{L}_{inv}(x_i) = -\log P_{v_i}(\mathcal{N}^h(i)|B(i)) \tag{4}$$

$$= -\log \frac{\sum_{p \in \mathcal{N}^h(i)} \exp(\bar{v}_p \cdot v_i/\tau)}{\sum_{n \in B(i)} \exp(\bar{v}_n \cdot v_i/\tau)} \tag{5}$$

The intuition behind the hard negative sampling is that it is not necessary to make the negative points that are already far from the anchor point be further, and it is more important to keep the ambiguous negative samples further. In practice, due to the inefficiency of computing the $B(i)$, we approximate $B(i)$ by $\mathcal{N}_M(i)$ and it works fairly well.

## 3.3 Overall Loss Function

At the begining of the training process, the discovered positive samples are not reliable due to the random network initialization. So we combine the Invariance Propagation loss with the instance discriminative loss $\mathcal{L}_{ins}$ together. For the instance discriminative term, the objective function with hard negative sampling is as follows:

$$\mathcal{L}_{ins}(x_i) = -\log P_{v_i}(i|\mathcal{N}_M(i) \cup \{i\}) \tag{6}$$

As the training proceeds, the discovered positive samples are more and more reliable. A reasonable strategy is to ramp up the weight of Invariance Propagation loss according to a time-dependent weighting function $\omega(t)$ as the training lasting. The total loss is formulated as:

$$\mathcal{L}(x_i) = \mathcal{L}_{ins}(x_i) + \lambda_{inv} \cdot \omega(t) \cdot \mathcal{L}_{inv}(x_i) \tag{7}$$

In practice, a simple binary ramp up function works well sufficiently. Specifically, we set $\omega(t)$ as 0 in the first $T$ epochs, i.e., $t \leq T$, and we set $\omega(t)$ as 1 when $t > T$.

Table 1: Linear classification results on ImageNet, Places205 and Pascal VOC07. We report 1-crop, top-1 accuracy. For ImageNet and VOC, we report the linear results for the output of the 16-th block. For Places205, we report the linear results for the output of the 15-th block. The BowNet has much more parameters than ordinary ResNet50 network because of the large size of fully connected layer.

| Method | Architecture | #Para | Epochs | ImageNet | Places | VOC |
|---|---|---|---|---|---|---|
| Supervised [28] | ResNet-50 | 26 | 200 | 75.9 | 51.5 | 87.5 |
| *Self-supervised learning methods* | | | | | | |
| Colorization [42] | ResNet-50 | 24 | 28 | 39.6 | 37.5 | 55.6 |
| Jigsaw [14] | ResNet-50 | 24 | 90 | 45.7 | 41.2 | 64.5 |
| Rotation [12] | ResNet-50 | 24 | 35 | 48.9 | 41.5 | 63.9 |
| BigBiGAN [9] | ResNet-50 | 24 | 488 | 56.6 | 49.8 | - |
| BoWNet(conv5) [11] | ResNet-50 | 65 | 280 | 60.5 | 50.1 | 78.4 |
| BoWNet(conv4) [11] | ResNet-50 | 65 | 280 | 62.1 | 51.1 | 79.3 |
| *Methods based on contrastive learning* | | | | | | |
| InsDis [39] | ResNet-50 | 24 | 200 | 54.0 | 45.5 | - |
| LocalAgg [46] | ResNet-50 | 24 | 200 | 58.8 | 49.1 | - |
| MoCo [16] | ResNet-50 | 24 | 200 | 60.6 | - | - |
| PIRL [28] | ResNet-50 | 24 | 800 | 63.6 | 49.8 | 81.1 |
| CMC [38] | ResNet-50-Lab | 47 | 400 | 64.1 | - | - |
| CPC [33] | ResNet-101 | 28 | - | 48.7 | - | - |
| CPC v2 [19] | ResNet-170 | 303 | - | 65.9 | - | - |
| AMDIM [1] | AMDIM | 626 | 150 | 68.1 | 55.0 | - |
| SimCLR [5] | ResNet-50-MLP | 28 | 1000 | 69.3 | - | 80.5 |
| MoCo v2 [6] | ResNet-50-MLP | 28 | 800 | 71.1 | - | - |
| PCL [26] | ResNet-50 | 24 | 200 | 62.2 | 49.2 | 82.2 |
| PCL [26] | ResNet-50-MLP | 28 | 200 | 65.9 | 49.8 | 84.0 |
| InvP (Ours) | ResNet-50 | 24 | 800 | 67.7 | 52.6 | 84.2 |
| InvP (Ours) | ResNet-50-MLP | 28 | 800 | **71.3** | **53.5** | **84.7** |

To efficiently retrieve the features of distractor samples to compute Eq 7, we maintain a memory bank to save features of all samples, which was proposed by Wu et al. [39]. Given the current calculated feature $v_i$ for $x_i$, we update the corresponding feature in memory bank as the exponential moving average of historical calculated features. The memory bank is initialized with random D-dimensional unit vectors and then update its values after each epoch, following the common design [39, 16, 46]. More details are in the supplementary material.

## 4 Experiments

In this section, we conduct quantitative and qualitative experiments to evaluate the proposed method. We train our unsupervised models on the training set of ImageNet [7]. We evaluate the quality of the learned representations on extensive downstream tasks, including linear classification on ImageNet, Places205 and Pascal VOC, semi-supervised classification on ImageNet, transfer learning on seven small scale datasets, and object detection. We also give the ablation study on several critical components of our method. We visualize the embedding statistics and the easy&hard positive neighbourhoods to provide qualitative analysis.

For all experiments, we set $\tau = 0.07$ for linear head and $\tau = 0.2$ for MLP head. We set $\lambda_{inv} = 0.6$, $T = 30$, $k = 4$, $M = 4096$, $l = 3$, $P = 50$. We use SGD optimizer with a momentum of 0.9 to optimize our models. The batch size is set to $128$ for ImageNet. More details can be found in the supplementary material.

### 4.1 Downstream Results

**Linear Classification.** We evaluate our method by linear classification on frozen features, following a common protocol proposed by [42]. Specifically, we freeze the parameters of convolutional layers, add a global average pooling layer, and train a linear classifier to classify images with true labels. We evaluate the linear classification results on all 17 blocks of ResNet-50 and report the best top-1, 1-crop

Table 2: Semi-supervised learning performance on ImageNet. We fine-tune our pre-trained models on 1% or 10% of ImageNet labeled data sampled from training set. We report top-5 accuracy on the held-out validation set. The results of other methods are adopted from original papers.

| Method | Architecture | Pretrain Epochs | Top5 Accuracy | |
|--------|--------------|-----------------|---------------|---|
| | | | 1% | 10% |
| *Semi-supervised learning methods* | | | | |
| VAT + Ent Min [15, 29] | ResNet-50v2 | - | 47.0 | 83.4 |
| S$^4$L Exemplar [41] | ResNet-50v2 | - | 47.0 | 83.7 |
| S$^4$L Rotation [41] | ResNet-50v2 | - | 53.4 | 83.8 |
| LLP [45] | ResNet-50 | - | 61.9 | 88.5 |
| *Unsupervised learning methods* | | | | |
| Jigsaw [14] | ResNet-50 | 90 | 45.3 | 79.3 |
| InsDis [39] | ResNet-50 | 200 | 39.2 | 77.4 |
| PIRL [28] | ResNet-50 | 800 | 57.2 | 83.8 |
| SimCLR [5] | ResNet-50-MLP | 1000 | 75.5 | 87.8 |
| PCL [26] | ResNet-50 | 200 | 75.6 | 86.2 |
| InvP (Ours) | ResNet-50 | 800 | 76.7 | 87.2 |
| InvP (Ours) | ResNet-50-MLP | 800 | **78.2** | **88.7** |

Table 3: Transfer learning performance on different datasets. We compare our method with SimCLR [5], supervised model and model trained from scratch. We report top-1 accuracy for CIFAR10, CIFAR100 and Stanford Cars; mean per-class accuracy for Caltech-101, Oxford-IIIT Pets and Oxford 102 Flowers; and the 11 point mAP for Pascal VOC2007, which is same as the setting of SimCLR

| Method | CIFAR10 | CIFAR100 | VOC | Caltech101 | Cars | Pets | Flowers |
|--------|---------|----------|-----|------------|------|------|---------|
| Scratch | 95.9 | 80.2 | 67.3 | 72.6 | 91.4 | 81.5 | 92.0 |
| Supervised | 97.5 | 86.4 | 85.0 | 93.3 | 92.1 | 92.1 | 97.6 |
| SimCLR | 97.7 | **85.9** | 84.1 | 92.1 | **91.3** | 89.2 | **97.0** |
| InvP (Ours) | **97.9** | 84.7 | **85.4** | **92.5** | 90.3 | **89.4** | 96.7 |

accuracy on the held-out evaluation set. The results of all 17 blocks are in the supplementary material. Table 1 shows the linear classification results on ImageNet, Places205 and Pascal VOC 2007 [10] respectively and gives the comparison with other state-of-the-art methods. Our method outperforms all others on all three datasets, closing the gap between supervised models and unsupervised ones on ImageNet and VOC2007, surpassing the supervised model on Places205 by a nontrivial margin of 2%, which shows the effectiveness of our method. We note that SimCLR [5] and MoCo [16, 6] concentrate on the improvement to the memory bank, which is orthogonal to our method. We believe that our method will achieve better results by replacing the memory bank with a momentum queue used in MoCo or directly calculating features with large batch size as SimCLR.

**Semi-supervised Learning.** We validate our method on the task of semi-supervised learning. We follow the setting of [41] to randomly choose 1% and 10% labeled images from ImageNet training set, and fine-tune the pre-trained unsupervised models. We report the top-5 accuracy on the held-out validation set. The results are shown in Table 2. Our method outperforms other semi-supervised and unsupervised methods, setting a new state-of-the-art on large scale semi-supervised classification task. With only 1% labeled images, our method achieves 78.2% top-5 accuracy, surpassing previous best result by an absolute margin of 2.6%, which shows the high quality of the learned features.

**Transfer Learning.** To investigate the transferability of our unsupervised models, we evaluate the fine-tuning performance of our method on seven different datasets. Specifically, we choose four natural image datasets: CIFAR10 and CIFAR100 [22], Pascal VOC2007 [10] and Caltech-101 [25], as well as three fine-grained classification datasets: Stanford Cars [21], Oxford-IIIT Pets [34] and Oxford 102 Flowers [30]. We first train the unsupervised model on ImageNet without labels. Then we fine-tune the pre-trained unsupervised model on the above seven datasets. In this experiment, we use the ResNet-50 with MLP projection head as the backbone. The results are shown in Table 3. Specifically, our method surpasses SimCLR on CIFAR10, VOC2007, Caltech101, and Pets. On the other three datasets, we also achieve competitive results. Besides, compared with SimCLR that

Table 4: The results of object detection. We fine-tune the unsupervised model on the Pascal VOC2007+2012 training set and report $AP_{50}$, $AP_{75}$ and $AP_{all}$ on VOC2007 test set, which is a widely adopted setting [16, 28, 14]. The proposed method outperforms other competitors.

| Method | Dataset | Network | AP | $AP_{50}$ | $AP_{75}$ |
|---|---|---|---|---|---|
| Supervised | ImageNet-1k | R50 C4 | 53.2 | 80.8 | 58.5 |
| Jigsaw [14, 28] | ImageNet-22k | R50-C4 | 48.9 | 75.1 | 52.9 |
| InsDis [39] | ImageNet-1k | R50-C4 | 52.3 | 79.1 | 56.9 |
| MoCo [16] | ImageNet-1k | R50-C4 | 55.2 | 81.4 | 61.2 |
| MoCo [16] | ImageNet-1k | R50-C5 | 53.8 | 81.1 | 58.6 |
| PIRL [28] | ImageNet-1k | R50-C4 | 54.0 | 80.7 | 59.7 |
| BoWNet [11] | ImageNet-1k | R50-C4 | 55.8 | 81.3 | 61.1 |
| MoCo v2 [6] | ImageNet-1k | R50-C4 | **57.4** | **82.5** | **64.0** |
| InvP (Ours) | ImageNet-1k | R50-C4 | 56.2 | 81.8 | 61.5 |

requires a large batch size of 4096 to allocate on 128 TPUs, our method is much easy to implement on standard hardware with only a batch size of 128.

**Object Detection.** We further evaluate the learned unsupervised models on object detection. Following [16], we fine-tune the learned unsupervised models (ResNet-50 with MLP) on Pascal VOC dataset [10], training on the VOC2007+2012 training set and evaluating on the VOC2007 test set. We use the Faster-RCNN-C4 object detector [36] with ResNet-50 as the backbone and report the detection performance in terms of $AP_{50}$, $AP_{75}$ and $AP_{all}$. The results are presented in Table 4. Our method outperforms most alternative competitors, including the ImageNet supervised one. We believe our method will get better detection results by replacing the memory bank with the momentum queue proposed in MoCo [16, 6].

Table 5: Results of ablation study on ImageNet linear classification. We train all models with 200 epochs and report the top-1 center-crop accuracy. The backbone is ResNet-50 without MLP head.

| | InvP | KNN | Without Hard Positive | Without Hard Negative |
|---|---|---|---|---|
| Acc | 63.3 | 57.6 | 60.7 | 61.9 |

## 4.2 Ablation Study

**Comparison with KNN**. We study the impact of our positive samples discovery algorithm. We implement an alternative method by using the KNN algorithm to find positive samples, which is similar to [20]. The ImageNet linear classification results are shown in Table 5, from which we can observe that our method outperforms the KNN counterpart by a large margin, which shows the effectiveness of our positive samples discovery algorithm. It is noted that we implement all models in Table 5 for 200 epochs using ResNet-50 with linear projection head as the backbone.

**Impact of Hard Positive Samples**. To investigate the effectiveness of the hard positive sampling strategy, we implement an alternative model by disabling the hard positive sampling strategy and train the model for 200 epochs. Table 5 shows the result. Without the hard sampling strategy, the linear classification performance decreases from 63.3 to 60.7. The results show the effectiveness of the hard positive sampling strategy. We believe this is because the hard positive samples provide more intra-class variations, which will be further analyzed in the following subsection.

**Impact of Hard Negative Samples.** Table 5 also shows the comparison between the model with the hard negative sampling strategy and the model without the hard negative sampling strategy. We observe that the hard negative sampling strategy gives an absolute improvement of 1.4% on ImageNet linear classification task, from 61.9% to 63.3%. The results show that concentrating on separating the ambiguous negative samples is more effective than separating all negative samples evenly.

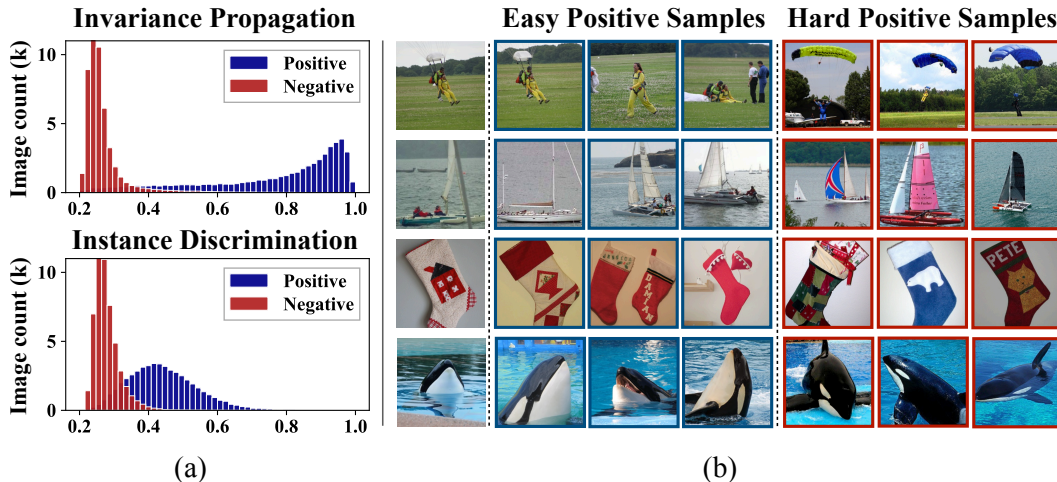

Figure 2: (a): The distribution of positive and negative similarities. We compare our model with the instance discrimination model [39]. (b): Visualization of the positive samples. The first column represents the anchor samples (query). For each anchor sample, we give comparison between the easy positive samples and the hard positive samples.

## 4.3 Analysis

**Embedding Statistics.** To understand the semantic properties of the learned representations, we calculate the 5 nearest neighbors from the same category as well as the 5 nearest neighbors from different categories [40] (We randomly sample 1500 images from different categories to make the positive and negative candidates balance). The distributions of similarities of the two settings are shown in Fig 2 (a). We compare our method with instance discrimination [39], which is a representative method to learn instance level invariant features. We have the following observations. (**1**) For samples from different categories, the similarity distributions of the two methods are similar. (**2**) For samples from the same category, our method tends to give them much higher similarities than instance discrimination. Most similarities of our method are concentrated on around 0.9, while the similarities of instance discrimination are concentrated on around 0.5. (**3**) For the overlap between positive and negative distributions, our method has only negligible overlap, while the instance discrimination has a relative large overlap. Overall the positive and negative similarity distribution of our model is more separable than the instance-wise method. We believe this is caused by the fact that our method considers the inter-instance relations to learn intra-class invariant representations so that for the positive samples from the same category, our method confidently gives higher similarities.

**Neighborhoods visualization.** To give an intuitive understanding of the hard positive sampling strategy, we display the easy&hard positive samples generated by our method in Fig 2 (b). We observe that the easy positive samples are more similar with anchor samples than hard positive samples in textures, colors, patterns, and views, which provide some low-level semantic unrelated information. For the hard positive samples, the semantic content is preserved while they provide more variations of the low-level information. For example, in the first row of Fig 2 (b), the hard positive samples present more colorful parachutes. In the second row, the hard positive samples present sailboats of different patterns and colors compared with the easy positive ones. For the last row, the easy positive samples are all about the head part of the dolphin, while the hard positive ones provide the whole body of different views of the dolphin. The intra-class variations brought by hard negative samples help the network learn more invariant representations.

## 5   Conclusion

In this paper, we propose Invariance Propagation, a novel unsupervised learning method to learn representations invariant to intra-class variations from large numbers of unlabeled images. It encourages all positive samples to reside in the same high-density region and a hard sampling strategy is used to provide more intra-class variations to help capture more abstract invariance. The learned

representations can be useful for a wide range of downstream tasks and extensive experiments are conducted on tasks of linear classification, semi-supervised learning and transfer learning. Both the quantitative and qualitative results demonstrate the superiority of the proposed method over state-of-the-art methods, and some results even surpass supervised models.

## Broader Impact

This work presents a novel unsupervised learning method, which effectively utilizes large numbers of unlabeled images to learn representation useful for a wide range of downstream tasks, such as image recognition, semi-supervised learning, object detection, etc. Without the labels annotated by humans, our method reduces the prejudice caused by human priors, which may guide the models to learn more intrinsic information. The learned representations may benefit robustness in many scenarios such as adversarial robustness, out-of-distribution detection, label corruptions, etc. What's more, the unsupervised learning can be applied to autonomous learning in robotics. The robot can autonomously collect the data without specifically labelling it and achieve lifelong learning.

There also exist some potential risks for our method. Unsupervised learning solely depends on the distribution of the data itself to discover the information. Therefore, the learned model may be vulnerable to data distributions. With biased dataset, the model is likely to learn incorrect causality information. For example, in the autonomous system, it is inevitable that the bias will be brought during the process of data collection due to the inherent constraints of the system. The model can also be easily attacked when the data used for training is contaminated intentionally. Additionally, since the learned representation can be used for a wide range of downstream tasks, it should be guaranteed that they are used for beneficial purposes.

We see the effectiveness and convenience of the proposed method, as well as the potential risks. To mitigate the risks associated with using unsupervised learning, we encourage the research to keep an eye on the distribution of the collected datasets and stop the use of the learned representations for harmful purposes.

## Acknowledgments and Disclosure of Funding

This work was supported in part by the National Key Research and Development Program under Grant 2018YFB1305102, and the Guoqiang Research Institute Project under Grant No. 2019GQG1010.

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
