[Supplementary Material]

# Unsupervised Representation Learning by Invariance Propagation: Supplementary Material

**Feng Wang, Huaping Liu, Di Guo, Fuchun Sun**
Department of Computer Science and Technology, Tsinghua University
`wang-f20@mails.tsinghua.edu.cn,hpliu@tsinghua.edu.cn`
`guodi.gd@gmail.com,fcsun@tsinghua.edu.cn`

## A  Additional Results

**Layer-wise Linear Results:** ResNet-50 consists of 17 residual blocks. We evaluate the quality of representations for each block. Specifically, for the output of each block, we add a global average pooling layer to map the output to about 9000-D vectors. Then, we freeze the parameters of all convolutional layer and train the linear layer on ImageNet [3] and Places205 [13]. Fig 1 shows the results. From Fig 1, we observe that the best results of ImageNet and Places205 linear classifications come from the 16-th block and 15-th block, respectively. Using the last block as output gives a slight performance decrease, which shows the last block may preserve more task-specific knowledge which is not suitable for downstream classification tasks.

Figure 1: Layer-wise linear classification results for ImageNet and Places205. We report the top-1 1-crop accuracy. For ImageNet, the best result comes from the 16-th block. For Places205, the best result comes from the 15-th block.

Table 1: Weighted-KNN Classifier Results on ImageNet.

| Method | InsDis [12] | LocalAgg [14] | MoCo [5, 6] | PCL [6] | InvP(Ours) |
|---|---|---|---|---|---|
| Accuracy | 46.5 | 49.4 | 47.1 | 54.5 | **61.3** |

**Weighted-KNN Classifier Results on ImageNet:** Following [12], we exhibit the weighted K-nearst neighbor classification results on ImageNet, which is another widely accepted criterion for contrastive learning based method. Specifically, for an image with feature $v_i$, we retrieve its top K nearest neighbors from the memory bank, and give a coefficient $exp(s_i/\tau)$ for the corresponding neighbor

according to [12] ($s_i$ is the similarity between $v_i$ and the corresponding neighbor, we set $\tau$ as 0.07), details can be found in [12]. Table 1 shows the K-NN results. Our method outperforms others by large margins.

**Embedding Visualization** We visualize the learned representation using t-SNE [10]. We choose the first 40 classes of ImageNet. For each class, we randomly choose 200 images. Fig 2 shows the results. We compare our method with Instance Discrimination [12]. It is obvious that our representations are more separable than the representations learned by Instance Discrimination, which shows that representations learned by our methods are more discriminative for semantic categories.

**(a)** InsDis Class 1-20

**(b)** InvP Class 1-20

**(c)** InsDis Class 20-40

**(d)** InvP Class 20-40

Figure 2: T-SNE visualization for the first 40 classes of ImageNet. We compare our method with the Instance Discrimination [12]. (a) and (c) are T-SNE visualization of representations learned from Instance Discrimination. (b) and (d) are T-SNE visualization of the representations learned from our method.

## B  Implementation Details

### B.1  Pretraining

We adopt a ResNet-50 as the backbone. For MLP projection head, we follow [1] to replace the last fully-connected layer by a two-layer MLP, which consists of a 2048-2048 fully-connected layer with a ReLU function, and a 2048-128 fully-connected layer. The output of the network is normalized by its L2-norm. The temperature $\tau$ is set to 0.07 for the model with linear projection head and 0.2 for the model with MLP projection head. The calculated features are saved to a memory bank following [12]. The settings of memory bank are same as [12] (The $m$ is set to 0.5). The data augmentation setting follows [12, 1, 2]: a $224 \times 224$-pixel crop is taken from a randomly resized image, and then undergoes random color jittering, random grayscale conversion, random gaussian blur [1, 2], and random horizontal flip. All the data augmentations are implemented by PyTorch [8].

For training the unsupervised models, we optimize our models with SGD optimizer with a momentum of 0.9. The batch size is set to 128 for ImageNet. We train our models for 800 epochs with a cosine learning rate schedule [7], and the initial learning rate is set to 0.03.

## B.2 Linear Classification

For linear classification, we freeze all convolutional layers of the learned ResNet-50. For ImageNet, we add a global average pooling layer for each block. For Places205 and Pascal VOC 2007, we add an average pooling layer for each block to pool the features to about 9000 dimensions. We add a linear layer and train it on ImageNet [3], Places205 [13] and Pascal VOC 2007 [4]. For training the linear model, we use an SGD optimizer with a momentum of 0.9. We train the models for 100 epochs with a batch size of 256 and set a decay rate of 10 for 60 and 80 epochs. The initial learning rate is set to 30.0 for ImageNet and 3.0 for Places205 and VOC2007. We set the weight decay as 0.0 for all datasets.

## B.3 Semi-supervised Learning

For ImageNet semi-supervised learning, we use the pre-trained model as initialization and randomly initialize the linear layer. We fine-tune all layers on subsets of ImageNet. Following [12, 1], we randomly choose 1% and 10% labels. For convolutional layers, we set a learning rate of 0.0012. For the last fully-connected layer, we set a learning rate of 0.06. For fine-tuning the model, we use an SGD optimizer with a momentum of 0.9. We train the models for 30 epochs with a batch size of 128 and set a decay rate of 10 for 15, 20 and 25 epochs. We set the weight decay as 1e-5.

## B.4 Transfer Learning

For transfer learning, we use the pre-trained models as initialization and fine-tune the model on other small scale datasets. We use SGD with momentum of 0.9. We train our models for 200 epochs using a batch size of 128 and set a decay rate of 10 for 120 and 160 epochs. We set the learning rate to 0.001 for all convolutional layers and 0.005 for the last fully-connected layer. We set the weight decay as 1e-6.

## B.5 Object Detection

For object detection, we use the Faster R-CNN [9] with the backbone of R50-C4. We fine-tune all layers end-to-end, following the setting in [5]. We fine-tune our model using the Detectron2 [11] framework with a learning rate of 0.005 (4 GPUs).