[Reviews · NeurIPS 2020]

Review 1

Summary and Contributions: This is an interesting work that shows that mining hard positives and hard negatives, found by measuring similarity via dot product between two vector representations, can improve over using the current instance as positive and uniformly sampling negatives. This is the main contribution of the paper along with experiments showing proposed negative and positive selection improve over other ways to choose positives and negatives for contrastive learning.

Strengths: This work shows improvements over the state of the art on a variety of image recognition tasks typically used to measure performance of self-supervised learning algorithms. The main mechanism of improvement is by mining hard positives and hard negatives during the self supervised portion of the training. The authors make a convincing case (via experimental results) that such a scheme does, in fact, improve the learned representations compared to other popular approaches such as MoCo and SimCLR.

Weaknesses: The work does not make it very clear how exactly the kNN graph (shown in figure 1) is actually constructed. Is it by computing pairwise distances between all samples using a similarity metric from equation 1? Is this the "graph distance" mentioned in the paper? Elaborating on this would go a long way to improve the quality of the paper. In addition, while many of these architectures are well known, it would be nice to have a rough description of the architecture used in the paper (at least the "mlp" part of resnet50-mlp). It would also be nice to have a more extensive ablation study to measure the impact of the a) number of positives/negatives to mine, b) what happens if instance positive loss component is disabled after invP loss component is enabled

Correctness: The claims and empirical methodology to evaluate their effectiveness appear correct, however as mentioned earlier greater clarity on actual construction of the graph used to select hard positives and negatives would help evaluate correctness of the claims beyond the empirical results.

Clarity: The paper is pretty well written, however there are a number of typos: line 126: "uses" lines 174-175 seem to have errors ("If not More") line 223 typo ("impact") line 251: easy&hard

Relation to Prior Work: This work should make a bit more effort to discuss differences with prior work, for example Deep(er) Cluster works use clustering methods for grouping negatives and positives, while other works have also mined hard positives and negatives (e.g. "Smart Mining for Deep Metric Learning")

Reproducibility: Yes

Additional Feedback: I've read the author rebuttal and thank the authors for their clarifications. I am believe my rating is still appropriate for this work.


Review 2

Summary and Contributions: This paper proposes a new self-supervised learning method, aiming to cluster the similar (positive) examples and pull away the dissimilar (negative) examples, in the representation space. The experimental results reveal the proposed approach is compelling.

Strengths: 1. Well written paper with clear intuition for the proposed methodology. 2. Noticable improvement from the previous work (for example [39]) 3. The results are pretty decent.

Weaknesses: With some theoritical analysis, the paper would be improved. But the extensive experiments and analysis certainly have made up for that.

Correctness: Yes.

Clarity: Yes

Relation to Prior Work: Yes

Reproducibility: Yes

Additional Feedback:


Review 3

Summary and Contributions: The paper proposes a invariance propagation approach for unsupervised learning of visual representations. It builds on the assumption that abstract embeddings which lie closely to each other should share high-level semantic labels. The method discovers positives samples in an iterative diffusion manner. The learning signal for training considers to contrast the hard positives and hard negative background samples. Improvements are observed in several benchmarks.

Strengths: * A novel clustering-based approach for unsupervised learning. It mines the nearest neighbors to form local clusters. * The mining approach which iteratively propagates similar metrics seems interesting. The diffusion is empirical and deserves more discussions with spectral methods for clustering.

Weaknesses: * While the paper presents a set of ablation experiments, it still lacks a great deal of analysis to explain the idea. For example, I am extremely curious about the accuracy for predicting the positive samples and negatives samples. How the quality of this invariance propagation affects the learning? * The paper builds on top of the InstDisc method. Since it is no longer a strong baseline, I am wondering can it be applied to MoCo as well? How does it likely to perform? * While the paper conducts a number of empirical experiments, it is not clear what the baseline approach that this approach should be compared to, and how much the improvement is. * The method seems to rely on a number hyper-parameters, k, l, P.

Correctness: I find no significant wrong claims in the paper.

Clarity: The paper is easy to follow and the graphics are good to understand.

Relation to Prior Work: * The paper lacks a thorough comparison and explanations against the local aggregation approach, which tries to solve a similar problem. Though the local aggregation paper is mentioned in the related works, no connections are discussed and how they are differentiated.

Reproducibility: Yes

Additional Feedback: My concerns are well addressed in the rebuttal.


Review 4

Summary and Contributions: This paper focuses on the unsupervised representation learning task. Different from previous image-level variations, the author focuses on the category-level variations. The proposed method recursively discovers semantically similar image samples as neighbors and tries to maximize the agreement between images from the same category. The results on the ImageNet classification and related downstream tasks look promising.

Strengths: 1. Category-level variations are more representative than image-level variations. 2. The hard sampling strategy for finding good positive and negative samples is reasonable and effective. 3. The evaluation results are good on both the classification task and the related downstream tasks.

Weaknesses: 1. The experimental comparisons are not enough. Some methods like MoCo and SimCLR also test the results with wider backbones like ResNet50 (2×) and ResNet50 (4×). It would be interesting to see the results of proposed InvP with these wider backbones. 2. Some methods use epochs and pretrain epochs as 200, while the reported InvP uses 800 epochs. What are the results of InvP with epochs as 200? It would be more clear after adding these results into the tables. 3. The proposed method adopts memory bank to update vi, as detailed in the beginning of Sec.3. What the results would be when adopting momentum queue and current batch of features? As the results of SimCLR and MoCo are better than InsDis, it would be nice to have those results.

Correctness: Yes

Clarity: The paper is well writen and well organized.

Relation to Prior Work: Yes

Reproducibility: Yes

Additional Feedback:

[Author Response · NeurIPS 2020]

We thank all reviewers for giving us the insightful comments.

**Relation to prior works (To reviewer #1 and #3)** We have stated the similarities between our method and other
relevant works in our paper (line 82-88). The differences between our work and other previous relevant category level
self-supervised methods lie in two aspects: **(1)** the way to generate positive samples. We use a novel positive sample
discovery method which is totally different from all other methods, such as Deep(er) Cluster, Local Aggregation, etc.
Our positive sample discovery algorithm makes the network obtain additional prior information in the training process,
i.e., the transitivity of semantic consistency. **(2)** the proposed hard sampling strategy. As far as we know, we are the first
to design hard sampling strategy in self-supervised contrastive learning. The above two differences are exactly the two
main components of our method.

**The way to construct kNN graph (To reviewer #1)** Indeed, the kNN graph is constructed by computing pairwise
similarities incrementally. For each iteration, we calculate the nearst k neighbours of each anchor sample and record the
indices of them. Then we collect all positive samples by a Breadth-First Search algorithm. This process is reflected as
Eq (3) and Figure 1 in our paper.

**Explanation of the Graph Distance (To reviewer #1)** The Graph Distance means the shortest path between two
vertices in the kNN Graph (the weight of each edge is set to 1). We use the graph distance to give another understanding
to our positive sample discovery algorithm, which is that the positive samples discovered by our algorithm can be
regarded as those samples whose graph distances from $v_i$ are less or equal than $l$ ($l$ is the layer to propagate).

**Other Ablations (To reviewer #1)** We have found that only using InvP loss (without instance loss) will result in low
speed of convergence, which is due to the unstable neighbours discovered by random initialized network. Besides,
only using InvP loss has the phenomenon of training oscillation. For the impact of the number of positive samples or
negatives samples, we have empirically tested several groups of hyperparameters, which shows that the number of
positive samples is not very sensitive when the value is between 30 and 70. We will add these results to our paper. We
thank review #1 for giving some detailed advices to improve the quality of our paper, such as the descriptions of some
architechtures and typos, we will adopt these advices to further improve the paper.

**The accuracy for predicting the positive and negatives samples (To reviewer #3)** For the accuracy of predicting
the positive samples and negatives samples. We have provided such statistics in Table 1 (in Supplementary Material)
and Figure 2 (a) (in our paper). Table 1 in Supplementary Material directly reflects the accuracy of predicting positive
samples. Concretely, using the nearst neighbours (can be regarded as the positive samples when the training process has
converged) to predict the label of each sample reaches an accuracy of 61.3%, surpassing other results, which shows the
high quality of the neighbours (positive samples). In Figure 2 (a), we also show the similarity distribution of positive
samples and negative samples, which can reflect the good quality of positive samples and negative samples discovered
by our algorithm. We also give the qualitative analysis in Figure 2 (b).

**Baseline (To reviewer #3)** About the baseline, our baseline is the KNN method, i.e. directly using the nearst $K$ samples
as positive samples. We have compared our algorithm with the KNN algorithm in Sec 4.2.

**Connection with MoCo (To reviewer #3 and #4)** As for the problem of MoCo, since we propose a general unsuper-
vised learning algorithm, our goal is to find more accurate and useful positive and negative samples to help the network
learn more useful features, while MoCo solves the problem of saving the sample features more effectively. Therefore,
they are actually two different and orthogonal problems. MoCo can be applied to the proposed method by replacing the
memory bank with the momentum queue (we pointed out this in line 187, page 6 of our paper). Due to the effectiveness
of the momentum queue, we believe a better performance can be obtained. However, the improvements are not caused
by our algorithm, and thus this setup does not reflect the performance of our method. In our paper, we use memory
bank because it is simple and efficient. By contrast, using momentum queue requires two feedforward passes which
makes the training time longer.

**About Hyper-parameters (To reviewer #3)** The proposed method relies on hyper-parameters such as $k,l,P$. We have
evaluated several groups of hyper-parameters. The contrastive learning often takes much more time to converge than
ordinary supervised learning. Therefore, it is hard to test all these hyper-parameters thoroughly. The current chosen
hyper-parameters have worked fairly well, and we believe a further hyper-parameter search might lead to better results.

**Wider Architechtures (To reviewer #4)** As demonstrated in many relevant works, wider networks such as
ResNet50(2x) or ResNet50(4x) can improve the downstream performance, and it will be interesting to see these
results. However, we think this improvement does not derive from the proposed algorithm directly. In this paper, we
mainly focus on evaluating the performance of the method itself.

**Results with 200 epochs (To reviewer #4)** For results with 200 epochs, we have reported in Table 5 with linear
projection head. It can be seen form Table 5 that with 200 epochs, our result still outperforms all other methods appeared
in Table 1 (200 epochs). We will add these results in Table 5 to Table 1 to give a more clear comparison.

[Meta-Review · NeurIPS 2020]

All the reviewers found the contribution of the paper significant, with solid experimental work. Rebuttal addressed a number of concerns from the reviewers. The main concern remaining is the lack of theoretical analysis -- but the experimental part is strong enough for acceptance.